# Empowering Users in Digital Privacy Management through Interactive LLM-Based Agents

**Bolun Sun**
SNF Agora Institute
Johns Hopkins University
Baltimore, MD 21218, USA
bsun26@jhu.edu

**Yifan Zhou**
Institute for Artificial Intelligence
University of Georgia
Athens, GA 30602, USA
yifan.zhou1@uga.edu

**Haiyun Jiang\***
School of Computer Science
Fudan University
Shanghai, China
haiyunjiangnlp@gmail.com

## Abstract

This paper presents a novel application of large language models (LLMs) to enhance user comprehension of privacy policies through an interactive dialogue agent. We demonstrate that LLMs significantly outperform traditional models in tasks like Data Practice Identification, Choice Identification, Policy Summarization, and Privacy Question Answering, setting new benchmarks in privacy policy analysis. Building on these findings, we introduce an innovative LLM-based agent that functions as an expert system for processing website privacy policies, guiding users through complex legal language without requiring them to pose specific questions. A user study with 100 participants showed that users assisted by the agent had higher comprehension levels (mean score of 2.6 out of 3 vs. 1.8 in the control group), reduced cognitive load (task difficulty ratings of 3.2 out of 10 vs. 7.8), increased confidence in managing privacy, and completed tasks in less time (5.5 minutes vs. 15.8 minutes). This work highlights the potential of LLM-based agents to transform user interaction with privacy policies, leading to more informed consent and empowering users in the digital services landscape.

## 1 Introduction

The pervasive collection and processing of personal data by online services have elevated privacy concerns in digital interactions (Vicario et al., 2019). To address these issues, websites and applications are legally mandated to publish privacy policies detailing practices related to data collection, usage, sharing, and protection. However, these policies are *notoriously difficult* for the average user to comprehend due to their *legal complexity* (McDonald & Cranor, 2008), *dense language, and considerable length*. Consequently, they are frequently ignored or misunderstood, undermining the principle of informed consent and exposing users to potential privacy risks. This disconnect poses significant challenges (Harkous et al., 2018): it impedes users' ability to make informed decisions about their data and complicates efforts by regulators and organizations to ensure transparency, enforce compliance, and build trust. As privacy regulations become more stringent and policies more intricate, innovative approaches are urgently needed to bridge this comprehension gap.

Recent advances in natural language processing (NLP), particularly through the development of large language models (LLMs), offer promising solutions to these challenges. LLMs, such as those based on GPT architectures, have demonstrated remarkable capabilities in understanding and generating human-like text across various domains, including legal documents. These models have been successfully applied to tasks such as information extraction, content summarization, and question

---

[1]* Haiyun Jiang is corresponding author.

answering, showing potential in automating the analysis and interpretation of complex legal texts like privacy policies.

In this paper, we present a novel application of LLMs to improve user interaction with website privacy policies by developing an AI agent based on GPT-4o-mini. Our research aims to *empower users by improving their ability to navigate and comprehend privacy agreements, thus fostering better control over their personal data*. The study is structured in two parts and the Figure 1 illustrates the overall workflow of the proposed approach in this paper.

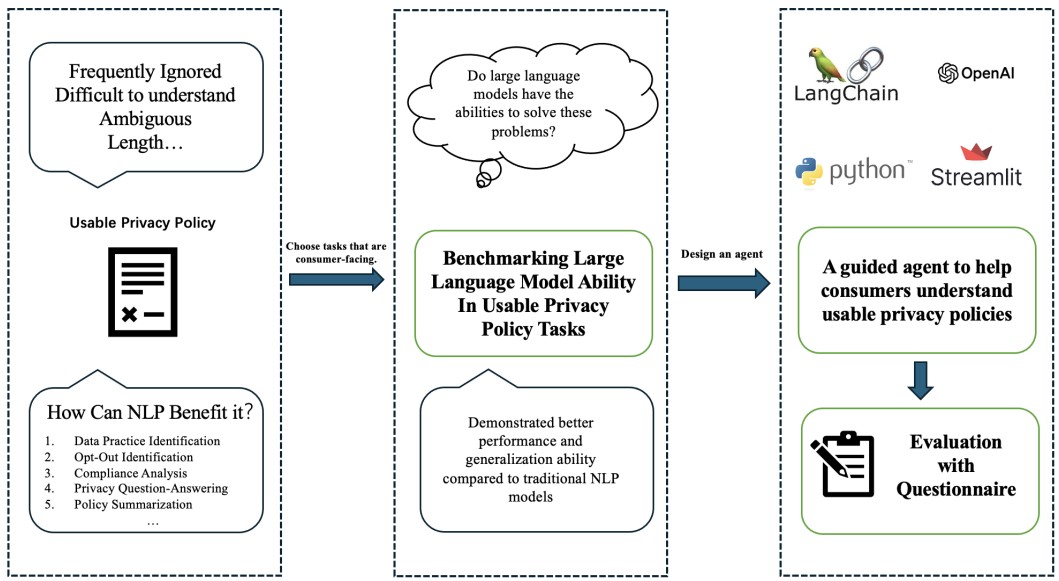

Figure 1: Overview of the workflow for benchmarking large language models on usable privacy policy tasks and designing a guided agent to assist consumers.

In the first part, we evaluate the performance of LLMs across several core tasks relevant to privacy policies, including Data Practice Identification, Choice Identification, Policy Summarization, and Privacy Question Answering (Wilson et al., 2016; Mysore Sathyendra et al., 2017; Ahmad et al., 2020; Keymanesh et al., 2020). By systematically replacing traditional models such as SVM, HMM, and CNN, as well as pre-trained models like BERT, with LLMs, we establish new benchmarks and demonstrate that *LLMs achieve state-of-the-art performance* across these tasks, significantly improving the *accuracy and efficiency* of natural language understanding, particularly in the analysis and interpretation of privacy policies.

Building on these empirical results, the second part introduces an innovative LLM-based agent built upon GPT-4o-mini to assist users in comprehending website privacy policies. The agent employs a heuristic interaction model that simplifies complex legal language without requiring users to know what specific questions to ask. As an expert system, it processes privacy agreements swiftly, executes multiple tasks such as identifying key information, and adapts to user needs by highlighting critical policy components autonomously (Ravichander et al., 2019).

To evaluate the effectiveness of our LLM-based agent, we conducted a user study involving 100 participants, divided into two groups: one interacted directly with privacy policies, and the other used the LLM agent for assistance (Kelley et al., 2009). We measured comprehension, efficiency, cognitive load, and user confidence. The results indicate that users who utilized the agent had a significantly better understanding of the privacy policy, reported reduced cognitive load, and completed tasks in less time compared to the control group. These findings demonstrate that our AI agent not only improves user comprehension but also enhances efficiency and boosts user confidence (Kelley et al., 2013).

The key contributions of this work are threefold:

- We provide empirical evidence of the *superiority* of LLMs over traditional models in interpreting privacy policies, setting new benchmarks.

- We develop an AI agent that *reduces cognitive load*, *increases user confidence*, and *enhances comprehension* of privacy policies. This agent simplifies complex legal language, assists users without requiring *prior knowledge* or *specific queries*, and autonomously highlights critical policy components to improve decision-making.

- We demonstrate that our LLM-based agent significantly improves users' ability to understand privacy policies, reducing task completion time and cognitive effort. This work not only *empowers users to better assert their privacy rights* and make more informed decisions about their data, but also provides valuable insights and a foundation for future research in developing AI-driven tools for legal document comprehension, encouraging further advancements in user-centered privacy solutions.

## 2 RELATED WORK

As privacy becomes a key concern in modern society, regulations like GDPR (General Data Protection Regulation) have shaped global privacy policies by enforcing stricter transparency requirements (Wang et al., 2018). Despite these advancements, privacy policies are still often long and complex, making them difficult for users to fully understand. McDonald and Cranor (McDonald & Cranor, 2008) estimated that reading all privacy policies a user encounters would take 201 hours per year, highlighting the need for more user-friendly formats.

**History of Privacy Policies Development**    The development of privacy rights began with Warren and Brandeis's concept of "the right to be let alone" (Warren & Brandeis, 1890), which laid the foundation for privacy as a legal right. The Universal Declaration of Human Rights (UN General Assembly, 1948) recognized privacy as a fundamental human right, while Westin (Westin, 1968) further expanded the concept, emphasizing individuals' control over personal information. As the digital age progressed, Introna (Introna, 1997) highlighted the importance of privacy in networked societies, and the Federal Trade Commission (Federal Trade Commission, 1998) introduced the "Notice and Choice" framework. Cate (Cate, 2010) criticized its effectiveness, arguing that it fails to adequately protect user privacy. More recent work has focused on the gap between the language of privacy policies and users' understanding (Reidenberg et al., 2015), and how big data and social inequalities affect privacy (Jain et al., 2016; Madden et al., 2017).

**Privacy Policies with Natural Language Models**    Natural Language Processing (NLP) has been instrumental in automating the analysis of privacy policies, addressing the increasing complexity of these documents. Early efforts by Costante et al. (Costante et al., 2012) and Ammar et al. (Ammar et al., 2012) focused on identifying data practices, while Liu et al. (Liu et al., 2014) and Ramanath et al. (Ramanath et al., 2014) worked on aligning privacy statements. The introduction of the OPP-115 dataset by Wilson et al. (Wilson et al., 2016) and the development of tools like Polisis (Harkous et al., 2018) advanced privacy policy classification and summarization. Machine learning techniques, such as CNNs for text classification and risk assessment tools (Zaeem et al., 2018), have improved the accessibility of privacy policies. Question-answering systems have also been applied to enhance user interaction with privacy policies, as shown by Ravichander et al. (Ravichander et al., 2019) and the creation of the PolicyQA dataset by Ahmad et al. (Ahmad et al., 2020).

**LLM Agent**    LLM Agents, powered by large language models like GPT and BERT, have shown significant promise in various fields, including social sciences and engineering (Bubeck et al., 2023). These agents excel in natural language understanding and interaction, making them ideal for automating tasks like policy summarization and compliance analysis. The application of LLM Agents to privacy policy analysis can enhance user accessibility, allowing for more efficient and understandable interpretations of complex documents.

# 3 BENCHMARKING LARGE LANGUAGE MODEL ABILITY IN USABLE PRIVACY POLICY TASKS

In this section, we present a comprehensive evaluation of OpenAI's state-of-the-art closed-source large language models (LLMs) on key tasks within the domain of usable privacy policies. The models assessed include GPT-4o to GPT-3.5. To accurately reflect the models' inherent capabilities and generalization potential, we employed both zero-shot and few-shot prompt engineering approaches, testing the models directly via their respective APIs. This methodology aligns with our objective of utilizing LLMs to develop agents capable of addressing a wide range of issues in privacy policy comprehension.

We selected the following four tasks for evaluation, as they represent critical aspects of user interaction with privacy policies: Data Practice Identification, Choice Identification, Policy Summarizing, Privacy Question Answering.

## 3.1 DATA PRACTICE IDENTIFICATION

In our experiments, we evaluated several models, including GPT-4o, GPT-4o-mini, GPT-4-turbo, and GPT-3.5, on the task of Data Practice Identification. The testing procedure involved direct API calls, employing the same prompt across all models to ensure consistency. These experiments were conducted on the entire OPP-115 dataset (Wilson et al., 2016), a comprehensive collection of annotated privacy policies.

Given that the task required the models to perform classification, we set the temperature parameter to zero to ensure deterministic outputs and eliminate randomness in predictions. This decision was made to maintain consistency and reliability in the classification process.

After considering factors such as response times, performance metrics, and computational cost, we selected GPT-4o-mini for further evaluation. Table 1, Table 2 present the performance of GPT-4o-mini on the Data Practice Identification task compared to the baseline(Wilson et al., 2016). Notably, GPT-4o-mini, under zero-shot learning conditions without additional context, outperformed the baseline model on average.

It is important to note that (Wilson et al., 2016) divided the *Other* category into three smaller subcategories, which are displayed in the final chart. Consequently, we do not have baseline data for the classification of the whole *Other* category. The results demonstrate the capability of GPT-4o-mini in policy classification tasks and its potential for application in constructing real-world agents.

| Category | GPT-4o-mini | | | LR | | |
|---|---|---|---|---|---|---|
| | Precision | Recall | F1-score | Precision | Recall | F1-score |
| First Party Collection/Use | 0.95 | 0.64 | 0.77 | 0.73 | 0.67 | 0.70 |
| Third Party Sharing/Collection | 0.84 | 0.69 | 0.75 | 0.64 | 0.63 | 0.63 |
| User Choice/Control | 0.88 | 0.43 | 0.58 | 0.45 | 0.62 | 0.52 |
| User Access, Edit, & Deletion | 0.90 | 0.59 | 0.71 | 0.47 | 0.71 | 0.57 |
| Data Retention | 0.96 | 0.16 | 0.27 | 0.10 | 0.35 | 0.16 |
| Data Security | 0.97 | 0.44 | 0.61 | 0.48 | 0.75 | 0.59 |
| Policy Change | 0.86 | 0.59 | 0.70 | 0.59 | 0.83 | 0.69 |
| Do Not Track | 0.64 | 0.88 | 0.74 | 0.45 | 1.00 | 0.62 |
| International & Specific Audiences | 0.95 | 0.77 | 0.88 | 0.49 | 0.69 | 0.57 |
| Other | 0.91 | 0.35 | 0.51 | NaN | NaN | NaN |
| Micro-Average | 0.90 | 0.53 | 0.67 | 0.53 | 0.65 | 0.58 |

Table 1: Performance metrics for GPT-4o-mini and LR across categories.

## 3.2 CHOICE IDENTIFICATION

We evaluated the performance of GPT-4o, GPT-4o-mini, GPT-4-turbo, and GPT-3.5 under the zero-shot learning setting without additional context on the Choice Identification task. The models exhibited performance slightly below the baseline but remained noteworthy. Specifically, we observed that precision was lower, while recall was relatively high. This suggests that the models were able

| Category | SVM | | | HMM | | |
|---|---|---|---|---|---|---|
| | Precision | Recall | F1-score | Precision | Recall | F1-score |
| First Party Collection/Use | 0.76 | 0.73 | 0.75 | 0.69 | 0.76 | 0.72 |
| Third Party Sharing/Collection | 0.67 | 0.73 | 0.07 | 0.63 | 0.61 | 0.62 |
| User Choice/Control | 0.65 | 0.58 | 0.61 | 0.47 | 0.33 | 0.39 |
| User Access, Edit, & Deletion | 0.67 | 0.56 | 0.61 | 0.48 | 0.42 | 0.45 |
| Data Retention | 0.12 | 0.12 | 0.12 | 0.08 | 0.12 | 0.09 |
| Data Security | 0.66 | 0.67 | 0.67 | 0.67 | 0.53 | 0.59 |
| Policy Change | 0.66 | 0.88 | 0.75 | 0.52 | 0.68 | 0.59 |
| Do Not Track | 1.00 | 1.00 | 1.00 | 0.45 | 0.40 | 0.41 |
| International & Specific Audiences | 0.70 | 0.70 | 0.70 | 0.67 | 0.66 | 0.66 |
| Other | NaN | NaN | NaN | NaN | NaN | NaN |
| Micro-Average | 0.66 | 0.66 | 0.66 | 0.60 | 0.59 | 0.60 |

Table 2: Performance metrics for SVM and HMM across categories.

to identify the majority of texts containing opt-out options, with minimal instances of missed true positives.

This trend indicates that the models rarely failed to detect texts that genuinely contained opt-out provisions. To further enhance performance, we employed a few-shot learning approach, deliberately increasing the number of examples with true opt-out options in the prompt. After this adjustment, the models achieved performance on par with, or slightly better than, the baseline.

The comparative results are illustrated in Table 3.

| Metric | GPT-4o-mini (zero shot) | GPT-4o-mini (few shots) | LR | BERT | fastText |
|---|---|---|---|---|---|
| Precision | 0.74 | 0.88 | 0.90 | 0.83 | 0.90 |
| Recall | 0.94 | 0.95 | 0.86 | 0.98 | 0.76 |
| F1-score | 0.83 | 0.91 | 0.88 | 0.90 | 0.82 |
| Accuracy | 0.94 | 0.93 | NaN | NaN | NaN |

Table 3: Performance metrics comparison across different models.

## 3.3 POLICY QUESTION ANSWER

In our experiments on the Privacy Question Answering task, we tested the performance of GPT-3.5 and GPT-4o-mini on the PolicyQA test dataset, designed to assess their ability to answer questions within the context of privacy policies. Additionally, we compared the performance of these models against the BERT-base model to further evaluate their effectiveness.

Our results indicate that the GPT-4o-mini model, utilizing a top-10 selection strategy, outperformed the BERT-base model in answering questions within the privacy policy context. The zero-shot performance of GPT-4o-mini demonstrates its ability to generalize across unseen data, effectively extracting relevant information and generating accurate answers from the privacy policies. However, we also observed that autoregressive large language models like GPT-4o-mini tend to hallucinate responses, even when strict output controls are in place. Despite our efforts to minimize such occurrences, hallucination remains a persistent challenge.

To address cases where the model failed to generate a meaningful response, we applied a postprocessing step. Specifically, we filtered out samples where the model provided no answer at all, ensuring that the final evaluation included only valid and relevant outputs. This step helped refine the model's overall performance assessment by removing uninformative outputs.

The comparative results are illustrated in Table 4.

| Metric | GPT-3.5 | GPT-3.5(top-10) | GPT-4o-mini | GPT-4o-mini(top-10) | BERT-base |
|--------|---------|-----------------|-------------|---------------------|-----------|
| Rouge-1 | 0.35 | 0.48 | 0.44 | 0.57 | 0.56 |
| Rouge-2 | 0.23 | 0.37 | 0.37 | 0.50 | 0.44 |
| Rouge-L | 0.32 | 0.45 | 0.43 | 0.56 | 0.56 |

Table 4: Performance metrics comparison for Policy Question Answer.

## 3.4 POLICY SUMMARIZATION

In this experiment, following the methodology of Keymanesh et al. (2020), we used their dataset to evaluate GPT-4o and GPT-4o-mini on ten publicly available user agreements from platforms like Google, Amazon, and CNN. These agreements were processed to extract the 'riskiest' sentences, focusing on privacy and data handling at content ratios of 1/16 and 1/64. While GPT-4o slightly underperformed compared to domain-specific supervised models, it showed strong generalization in a zero-shot setting. The dataset's age and lack of task-specific fine-tuning likely contributed to the performance difference, but GPT-4o's ability to achieve comparable results demonstrates its robust generalization.

| Metric | Compression Ratio 1/64 | | | | Compression Ratio 1/16 | | | |
|--------|--------|-------------|--------|--------|--------|-------------|--------|--------|
| | GPT-4o | GPT-4o-mini | CNN+RF | CNN+CF | GPT-4o | GPT-4o-mini | CNN+RF | CNN+CF |
| ROUGE-1 | 0.338 | 0.335 | 0.340 | 0.379 | 0.429 | 0.427 | 0.431 | 0.404 |
| ROUGE-2 | 0.248 | 0.246 | 0.250 | 0.288 | 0.310 | 0.308 | 0.312 | 0.287 |
| ROUGE-L | 0.246 | 0.244 | 0.248 | 0.292 | 0.366 | 0.364 | 0.368 | 0.340 |
| METEOR | 0.398 | 0.396 | 0.400 | 0.439 | 0.418 | 0.416 | 0.420 | 0.416 |

Table 5: Performance metrics comparison for different compression ratios.

## 4 LLM AGENT FOR USABLE PRIVACY POLICY

### 4.1 OVERVIEW OF THE AGENT

We designed the AI agent as a guided and heuristic system to assist users in comprehending complex privacy policies without requiring expertise in privacy law. It proactively identifies critical points that warrant user attention and employs guided, heuristic dialogues to effectively communicate this information(Ouyang et al., 2022; OpenAI, 2023; Wei et al., 2022). Subsequently, it facilitates an open-ended question-answering session to help users address any remaining uncertainties.

As illustrated in Figure 2, the agent operates through a multi-stage process, beginning with the retrieval of the privacy policy from a specified URL, followed by document preprocessing, segmentation, summarization, data practice identification, opt-out choice extraction, and question answering. The system autonomously recognizes essential sections—such as data-sharing practices, user rights, and opt-out mechanisms—that are often obscured within extensive legal texts. Additionally, it features an interactive interface that enables users to engage with the policy by posing specific inquiries regarding their privacy rights and options. By simplifying the process of navigating and understanding privacy policies, the AI agent enables users to make informed decisions about their privacy with ease and confidence. It provides clarity and support, allowing users to focus on what matters most without the need for extensive legal knowledge.

### 4.2 SYSTEM ARCHITECTURE

The AI agent is built upon the LangChain framework (Følstad & Skjuve, 2019), integrating large language models with specialized tools for privacy policy analysis. The architecture comprises the following key components:

**Document Retrieval and Preprocessing:** The agent accepts a user-provided URL to retrieve the corresponding privacy policy. Utilizing LangChain's `request_url`, it extracts relevant content, filtering out non-essential elements like advertisements to focus on the policy text.

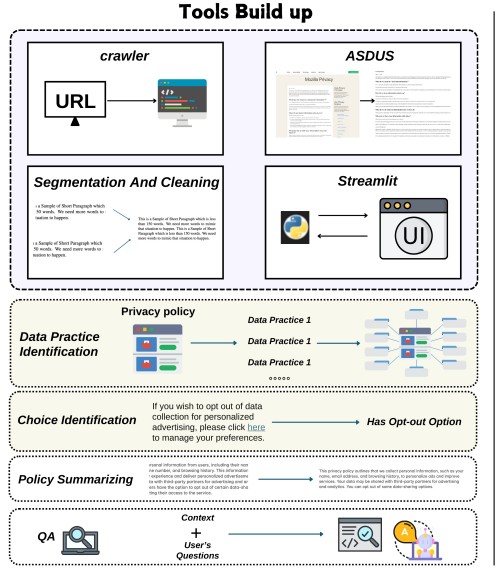
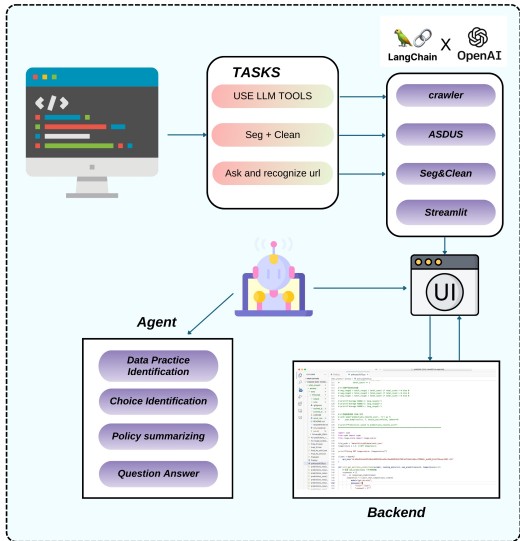

Figure 2: The figure illustrates the development workflow of this agent. The system integrates multiple tools to automate the extraction and interpretation of privacy practices, providing users with summarized insights and question-answering capabilities.

**Document Segmentation:** We employ ASDUS (Athreya, 2018) to segment the HTML document into titles and prose, organizing content into `<h2>` and `<p>` tags. This process aids in identifying critical sections such as user rights, data-sharing practices, and opt-out options, enhancing the agent's ability to analyze and interpret the policy effectively.

**Policy Analysis Modules:** The agent incorporates two primary modules: - *Policy Practice Identification:* Using LangChain's custom tools, the agent classifies policy segments into ten predefined categories (e.g., data collection, third-party sharing) based on established taxonomies (Wilson et al., 2016). - *Content Summarization:* The `summarize_tool` condenses complex policy content into concise summaries, allowing users to quickly grasp essential privacy practices without sifting through dense legal text.

**Opt-Out Choice Detection:** To detect opt-out options, the agent scans the policy text for relevant keywords and hyperlinks leading to opt-out mechanisms. It then uses the underlying LLM to analyze the context and confirm valid opt-out choices, which are stored for easy user access.

**Interactive Dialogue Mechanism:** The agent's interactive capabilities are facilitated by LangChain's `RunnableWithMessageHistory`, enabling dynamic question-and-answer sessions. Users can inquire about specific aspects of the policy, and the agent responds by leveraging appropriate tools and the LLM for contextually relevant answers. Conversation history is maintained to provide a coherent and personalized interaction experience.

**Backend Integration:** Managed by LangChain's `AgentExecutor`, the system orchestrates the execution of multiple tools efficiently. Integration with the underlying LLM (e.g., GPT-4o-mini) allows for deep contextual understanding and accurate response generation. The agent's architecture is designed for scalability, enabling parallel processing of multiple queries while ensuring robustness through memory databases and external modules for advanced tasks.

## 4.3 DATA SECURITY IN AGENT

Ensuring robust data security is paramount in privacy-oriented systems. Drawing on established best practices in the literature (He et al., 2018), our design adopts the principle of data minimization: user interactions with our agent neither require nor store personally identifiable information (PII).

Specifically, the system only processes publicly available privacy policies supplied by users and returns structured, human-readable explanations without retaining input or output logs.

Our system relies on the OpenAI API for natural language processing tasks, but does not cache any user queries or responses in a persistent database. By integrating these design choices—secure communications, ephemeral processing, and strict non-retention of user provided information we align with privacy by design guidelines and maintain the confidentiality of user interactions. This approach is well-suited to privacy research settings, as it ensures that users' sensitive data are neither collected nor exposed during the policy comprehension process.

## 5    QUESTIONNAIRE DESIGN

We designed a questionnaire to evaluate participants across five dimensions—comprehension, user experience, time efficiency, cognitive load, and trust intention—based on established evaluation methods (Brooke, 1996; Hart & Staveland, 1988; McKnight & Chervany, 2002). This allowed us to assess differences between participants who read privacy policies directly and those assisted by our AI agent. The complete questionnaire content is attached in the appendix.

**Comprehension Assessment:** Comprehension was assessed using three multiple-choice questions on key aspects of privacy policies: data collection types, data sharing practices, and user rights (Reidenberg et al., 2016). Participants scored 1 point for each correct answer, with a total possible score ranging from 0 to 3.

**User Experience Evaluation:** We evaluated user experience using a 5-point Likert scale to measure ease of use, satisfaction, and information quality (Brooke, 1996). Participants interacting with the AI agent answered additional AI-specific questions.

**Time Efficiency:** We recorded the time each participant took to complete the task—either reading the privacy policy or using the AI agent—to compare time efficiency between the two groups.

**Cognitive Load Evaluation:** Cognitive load was assessed using an adapted NASA-TLX questionnaire (Hart & Staveland, 1988). Participants rated mental demand, task difficulty, and frustration on a scale from 0 ("Very Low") to 10 ("Very High").

**Trust and Intention Assessment:** Participants rated their trust in the information received and their intention to use similar tools in the future using a 5-point Likert scale (McKnight & Chervany, 2002).

**Open-ended Feedback:** We collected open-ended feedback on any difficulties encountered and suggestions for improving the reading experience or the AI agent, providing qualitative insights for future enhancements.

## 6    RESULT ANALYSIS

This section presents the analysis of data collected from participants who directly read the privacy policies (*Control Group*) and those who used the AI agent (*Experimental Group*). We evaluated the results across five dimensions: comprehension, user experience, time efficiency, cognitive load, and trust/intention.

### 6.1    PARTICIPANTS AND DATA ANALYSIS

A total of 100 participants were recruited for the study, comprising 52 males and 48 females, ranging in age from 18 to 65 years old (mean age = 35.2, SD = 10.4). Participants were randomly assigned to the Control Group or the Experimental Group, with 50 participants in each. The sample included individuals with diverse educational backgrounds: 20% had a high school diploma, 50% held a bachelor's degree, and 30% had a graduate degree (master's or doctoral). Regarding digital literacy, participants self-reported their proficiency levels, with 45% identifying as beginners, 45% as intermediate users, and 10% as advanced users. This diversity in demographics ensures that the findings are generalizable across different user profiles.

Prior to analysis, we verified that the data met the assumptions for parametric tests, including independence of observations, normality, and homogeneity of variances using Shapiro-Wilk and Levene's tests. All assumptions were satisfied, allowing for the use of independent samples $t$-tests.

| Dimension | Measure | Control Group | Experimental Group |
|---|---|---|---|
| **Comprehension** | Mean Score (0–3) | 1.8 (0.7) | 2.6 (0.6) |
| | $t$-test ($p$-value) | $t(98) = 6.23, p < 0.001, d = 1.24$ | |
| **User Experience** (1–5) | Ease of Use | 2.8 (0.9) | 4.1 (0.7) |
| | Info Accessibility | 2.5 (1.0) | 4.2 (0.6) |
| | Info Clarity | 2.9 (0.8) | 4.3 (0.6) |
| | $t$-tests ($p$-value) | $p < 0.001, d = 1.69$–$2.13$ | |
| **Time Efficiency** | Time (minutes) | 15.8 (2.5) | 5.5 (1.8) |
| | $t$-test ($p$-value) | $t(98) = 11.62, p < 0.001, d = 2.32$ | |
| **Cognitive Load** (0–10) | Mental Demand | 7.5 (1.2) | 3.5 (1.1) |
| | Task Difficulty | 7.8 (1.1) | 3.2 (1.0) |
| | Frustration Level | 7.2 (1.3) | 3.0 (1.2) |
| | $t$-tests ($p$-value) | $p < 0.001, d = 2.23$–$2.81$ | |
| **Trust and Intention** (1–5) | Trust in Info | 2.8 (0.7) | 4.5 (0.5) |
| | Confidence in Privacy Mgmt | 2.6 (0.8) | 4.4 (0.6) |
| | $t$-tests ($p$-value) | $p < 0.001, d = 1.57$–$1.62$ | |

Table 6: Summary of Results Across Measured Dimensions

The detailed analysis of each measured dimension is summarized below, in conjunction with the results presented in Table 6.1.

- **Comprehension**: Participants in the Experimental Group demonstrated significantly better comprehension of the privacy policies compared to the Control Group. An independent samples $t$-test confirmed this difference ($t(98) = 6.23, p < 0.001$, Cohen's $d = 1.24$).

- **User Experience**: The Experimental Group reported a significantly better user experience across all measured statements. Differences in ease of use, information accessibility, and information clarity were all statistically significant with large effect sizes (all $p < 0.001$, $d$ ranging from 1.69 to 2.13).

- **Time Efficiency**: Using the AI agent significantly reduced the time required to complete the task. The difference was statistically significant ($t(98) = 11.62, p < 0.001$, Cohen's $d = 2.32$), demonstrating the AI agent's efficiency.

- **Cognitive Load**: Participants in the Experimental Group experienced significantly lower cognitive load across all dimensions. Reductions in mental demand, task difficulty, and frustration level were statistically significant (all $p < 0.001$, $d$ ranging from 2.23 to 2.81), indicating the AI agent effectively reduced cognitive effort.

- **Trust and Intention**: The Experimental Group reported higher levels of trust in the information received and greater confidence in managing their privacy. These differences were statistically significant with large effect sizes (all $p < 0.001$, $d$ ranging from 1.57 to 1.62). Participants also expressed a strong intention to use the AI agent for understanding future privacy policies.

## 7 DISCUSSION & LIMITATIONS

While our study demonstrates the potential of LLM-based agents in enhancing user comprehension of privacy policies, it is important to acknowledge the limitations and consider areas for future work. One significant concern is the possibility of biases and hallucinations inherent in LLMs (Bender et al., 2021), which could affect the accuracy and reliability of the information provided to users. These issues may inadvertently mislead users or reinforce existing biases, potentially impacting their understanding and decisions regarding privacy. Addressing these concerns requires implementing

robust validation mechanisms and incorporating human oversight to ensure the agent's outputs are trustworthy.

Additionally, our study focused on short-term interactions with the agent. To fully understand its impact on users' privacy management behaviors, longitudinal studies are necessary. Evaluating the agent's effectiveness over extended periods will provide insights into its sustainability and long-term benefits, including whether users continue to engage with the agent and how it influences their privacy decisions over time. Future research should also explore user retention rates, the agent's impact on long-term privacy awareness, and its integration into daily digital practices.

## 8 CONCLUSION

In this work, we have pioneered the application of large language models to enhance user comprehension of privacy policies through an interactive LLM-based agent. We investigated the challenges users face when interpreting complex legal language and demonstrated how our agent significantly improves understanding, reduces cognitive load, and increases confidence in managing personal data. Simultaneously, we demonstrated new benchmarks in privacy policy analysis tasks, showcasing the superior performance of LLMs over traditional models. We strongly encourage further research in this area, including the development of more advanced agents and evaluation methods. By fostering a collaborative and iterative approach to user empowerment and privacy management, we eagerly anticipate continued advancements in promoting informed consent and user autonomy in the digital landscape.

## 9 ACKNOWLEDGMENTS

We would like to clarify that this research did not require formal IRB approval, as it involved no collection of personally identifiable information (PII) and presented minimal risk to participants. The study focused exclusively on comprehension tasks related to publicly available privacy policies, without requesting sensitive information or posing risks beyond those of everyday activities.

Since this was a personal research endeavor without organizational sponsorship, we followed ethical guidelines for minimal-risk research. Informed consent was obtained from all participants, who were fully informed about the study's purpose, the nature of the tasks, and their right to withdraw at any time. No personal or sensitive data were gathered or stored, ensuring that the project remained strictly centered on analyzing privacy policy content.

Participants were recruited on a voluntary basis through social platforms and personal networks. They received appropriate compensation for their time, ensuring fairness and ethical engagement. By adhering to these measures, we maintained a high standard of ethical conduct and respected the confidentiality of all individuals involved.

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

## A    APPENDIX

### A.1    PROMPT USED IN BENCH-MARKING AND AGENT

#### A.1.1    PRACTICE IDENTIFICATION

> **Prompt**
>
> Instruction: You are given an annotation scheme for a website's privacy policy, which consists of the following ten categories:
>
> 1. **First Party Collection/Use**: How and why the service provider collects user information.
> 2. **Third Party Sharing/Collection**: How user information may be shared with or collected by third parties.
> 3. **User Choice/Control**: What choices and control options are available to users.
> 4. **User Access, Edit, & Deletion**: Whether and how users may access, edit, or delete their information.
> 5. **Data Retention**: How long user information is stored.
> 6. **Data Security**: How user information is protected.
> 7. **Policy Change**: Whether and how users will be informed about changes to the privacy policy.
> 8. **Do Not Track**: Whether and how Do Not Track signals for online tracking and advertising are honored.
> 9. **International & Specific Audiences**: Practices that pertain to specific groups of users (e.g., children, Europeans, California residents).
> 10. **Other**: Additional labels for introductory or general text, contact information, or practices not covered by other categories.
>
> For the privacy policy text content below, please select the most appropriate category (by number) and return only the number.
>
> **Content:** ⟨Your Text Here⟩
>
> Answer:

#### A.1.2    POLICY SUMMARIZATION

> **Prompt**
>
> Instruction: Please select the **6** sentences from ⟨**TEXT**⟩ that you consider to be the most ⟨**RISKY**⟩ from the provided privacy policy text. You must strictly choose only sentences from the original text without adding, modifying, or including any other content.
> ⟨**RISKY**⟩ refers to sections or clauses that could potentially expose users to privacy threats, data misuse, or security vulnerabilities.
>
> ⟨**TEXT**⟩:

---

**Prompt**

Instruction: Please select the **29** sentences from ⟨**TEXT**⟩ that you consider to be the most ⟨**RISKY**⟩ from the provided privacy policy text. You must strictly choose only sentences from the original text without adding, modifying, or including any other content.
⟨**RISKY**⟩ refers to sections or clauses that could potentially expose users to privacy threats, data misuse, or security vulnerabilities.

⟨**TEXT**⟩:

---

### A.1.3   OPT-OUT CHOICE IDENTIFICATION

---

**Prompt**

Instruction: You are an intelligent assistant trained to identify whether a hyperlink in website privacy policies provides an opt-out option for users. Your task is to determine whether a hyperlink offers users the ability to withdraw consent for data collection or processing ('opt-out'). When making a decision, follow these stricter steps:

1. Review the context of the hyperlink carefully. Check if the link explicitly refers to an option for users to decline, refuse, or stop data collection or usage. The link should offer clear action to withdraw consent or change data preferences.
2. Look for specific keywords or phrases like 'opt-out,' 'unsubscribe,' 'do not sell,' 'withdraw consent,' 'manage preferences,' or 'disable tracking.'
3. If the hyperlink refers to generic terms such as 'privacy policy,' 'learn more,' 'terms of service,' 'support,' or 'about us,' return 'False.'
4. Be very cautious in interpreting implicit meanings. If there is any doubt about whether the link provides an opt-out action, return 'False.'
5. Only return 'True' if the link explicitly offers an opt-out or similar function directly related to data privacy or user preferences.

Here is the content to analyze. Please predict whether the hyperlink contains an opt-out choice based on the following information. Return only 'True' or 'False.'

**Content:** ⟨Your Content Here⟩

Answer:

---

### A.1.4 POLICY QA

> **Prompt**
>
> You are an expert in privacy policies. I will provide you with ⟨**Reading Material**⟩ and a ⟨**Question**⟩.
> Your task is to extract the **most precise and relevant information** from the ⟨**Reading Material**⟩ that answers the ⟨**Question**⟩.
>
> **Guidelines:**
> 1. **Only** use information directly from the ⟨**Reading Material**⟩ to generate your response.
> 2. If the relevant information is a specific word, phrase, or sentence, use **only that**. Avoid unnecessary elaboration.
> 3. If the exact answer is not present in the reading material, return the closest **direct** match in the ⟨**Reading Material**⟩.
>
> **Output Requirements:**
> 1. The Output must consist **only** of words or sentences that appear in the ⟨**Reading Material**⟩.

## A.2 AGENT USAGE EXAMPLES

To illustrate the practical application of our LLM-based agent, we provide screenshots demonstrating its interaction with users.

### A.2.1 USER INTERFACE

# Privacy Policy Analysis Agent

Hello, I'm your usable privacy policy assistant. I'm here to help you quickly understand the content of this privacy policy, please enter the URL of the Privacy Policy:

Figure 3: The initial screen of the Privacy Policy Analysis Agent, prompting users to enter a URL for detailed policy analysis.

A.2.2  DATA PRACTICES IDENTIFICATION

# Privacy Policy Analysis Agent

Hello, I'm your usable privacy policy assistant. I'm here to help you quickly understand the content of this privacy policy, please enter the URL of the Privacy Policy:

https://www.imdb.com/privacy

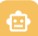 The IMDb Privacy Notice contains the following information:

1. **First Party Collection/Use**: 16 paragraphs detailing how and why IMDb collects user information.
2. **Third Party Sharing/Collection**: 11 paragraphs explaining how user information may be shared with or collected by third parties.
3. **User Choice/Control**: 11 paragraphs outlining the choices and control options available to users.
4. **User Access, Edit, & Deletion**: 4 paragraphs on if and how users may access, edit, or delete their information.
5. **Data Retention**: 1 paragraph on how long user information is stored.
6. **Data Security**: 2 paragraphs on how user information is protected.
7. **Policy Change**: 2 paragraphs on if and how users will be informed about changes to the privacy policy.
8. **Do Not Track**: 0 paragraphs on if and how Do Not Track signals for online tracking and advertising are honored.
9. **International & Specific Audiences**: 4 paragraphs on practices that pertain only to specific groups of users (e.g., children, Europeans, or California residents).
10. **Other**: 3 paragraphs for introductory or general text, contact information, and practices not covered by the other categories.

Additionally, there are 17 opt-out options available for users.

Ask a question about the privacy policy:    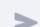

Figure 4: Comprehensive breakdown of the IMDb Privacy Notice, showing various sections and the extent of information provided.

### A.2.3 THIRD-PARTY SHARING/COLLECTION

Tell me more about the Third Party Sharing/Collection

The IMDb Privacy Notice provides detailed information about Third Party Sharing/Collection in the following ways:

1. **Transactions Involving Third Parties**: IMDb makes certain services available through third parties, such as purchasing movie tickets. In these cases, user information related to the transactions is shared with the third party involved.

2. **Third-Party Service Providers**: IMDb employs other companies and individuals to perform functions on its behalf, such as processing payments, analyzing data, and providing marketing assistance. These service providers have access to personal information needed to perform their functions but are not permitted to use it for other purposes.

3. **Business Transfers**: In the event that IMDb or its assets are acquired or merged with another company, user information may be transferred as part of the business assets. However, the information remains subject to the promises made in any pre-existing Privacy Notice unless the user consents otherwise.

4. **Protection of IMDb and Others**: IMDb may release account and other personal information to comply with the law, enforce agreements, or protect the rights, property, or safety of IMDb, its users, or others. This includes exchanging information with other companies and organizations for fraud protection and credit risk reduction.

5. **Third-Party Advertising and Links**: IMDb Services may include third-party advertising and links to other websites and apps. Third-party advertising partners may collect information about users when they interact with their content, advertising, and services.

6. **Use of Third-Party Advertising Services**: IMDb provides ad-serving companies, ad networks, and advertising companies with information that allows them to serve more relevant ads and measure their effectiveness. However, IMDb does not share users' names or other information that directly identifies them.

These practices ensure that while user information may be shared with third parties, it is done in a controlled manner with specific purposes and limitations.

Ask a question about the privacy policy: ➤

Figure 5: Detailed explanation of the third-party sharing and collection practices outlined in the IMDb Privacy Notice.

A.2.4    OPT-OUT DETECTION

Figure 6: An overview of the privacy policy analysis agent interface showing the opt-out options available to users.

### A.2.5  Policy Summarization

Summarize the policy in one paragraph and keep all the risky and most important information for me

The IMDb Privacy Notice outlines how personal information is collected, used, and shared through its services. It collects information provided by users, automatically through cookies, and from third-party sources to enhance services, personalize experiences, and comply with legal obligations. User data may be shared with Amazon, third-party service providers, and during business transfers, but IMDb does not sell personal information. Users have control over their data through account settings, including opting out of interest-based ads and managing cookies. IMDb employs security measures to protect user data but advises users to safeguard their account credentials. The policy also addresses data retention, user rights to access and delete data, and specific provisions for European and UK users under GDPR. Notably, IMDb services are not intended for children under 13, and users are informed of any significant policy changes.

Ask a question about the privacy policy:

Figure 7: A concise summary of the IMDb Privacy Notice, highlighting key aspects of data collection, sharing, and user control.

### A.3 USER PRIVACY POLICY READING EXPERIENCE QUESTIONNAIRE

BASIC INFORMATION

    1. Have you ever read a privacy policy?        [   ] Yes        [   ] No

    2. Have you ever used an AI agent?        [   ] Yes        [   ] No

EXPERIMENTAL GROUP

Please select the group you participated in:

    [   ] Group A: Directly read the privacy policy

    [   ] Group B: Used an AI agent to assist in reading the privacy policy

### I. COMPREHENSION ASSESSMENT

Based on your understanding after reading the privacy policy, please answer the following questions:

1. **Types of Data Collected:** What types of personal data are collected in this privacy policy?
   - (A) Contact information (e.g., email, phone)
   - (B) Location information
   - (C) Browsing history
   - (D) All of the above
2. **Data Sharing:** Does the company share your personal data with third parties?
   - (A) Yes
   - (B) No
   - (C) Only with user consent
   - (D) Uncertain
3. **User Rights:** According to the privacy policy, what rights do you have to manage your personal data?
   - (A) Access and modify your data
   - (B) Request deletion of your data
   - (C) Opt-out of data processing
   - (D) All of the above

### II. USER EXPERIENCE ASSESSMENT

Based on your experience, please rate the following statements:

1 = Strongly Disagree    2 = Disagree    3 = Neutral    4 = Agree    5 = Strongly Agree

1. **Usability**
   - a) I found reading/using the privacy policy/AI agent very easy.  1 [   ] 2 [   ] 3 [   ] 4 [   ] 5 [   ]
   - b) I could easily find the information I needed.    1 [   ] 2 [   ] 3 [   ] 4 [   ] 5 [   ]
   - c) The navigation and operation were intuitive to me.  1 [   ] 2 [   ] 3 [   ] 4 [   ] 5 [   ]
2. **Satisfaction**
   - a) I am satisfied with my understanding of the privacy policy. 1 [   ] 2 [   ] 3 [   ] 4 [   ] 5 [   ]
   - b) I am satisfied with the overall reading/usage experience.  1 [   ] 2 [   ] 3 [   ] 4 [   ] 5 [   ]
3. **Information Quality**

a) I believe the information provided was clear and useful.  1 [   ] 2 [   ] 3 [   ] 4 [   ] 5 [   ]

b) The content helped me better understand my privacy rights.   1 [   ] 2 [   ] 3 [   ] 4 [   ] 5 [   ]

4. **AI Agent Specific (Group B only)**

a) The AI agent's responses were accurate and relevant.   1 [   ] 2 [   ] 3 [   ] 4 [   ] 5 [   ]

b) I found interacting with the AI agent pleasant.      1 [   ] 2 [   ] 3 [   ] 4 [   ] 5 [   ]

c) I am willing to continue using AI agents to understand privacy policies in the future.  1 [   ] 2 [   ] 3 [   ] 4 [   ] 5 [   ]

## III. TIME EFFICIENCY

Please indicate the total time you spent completing the task (in minutes): _________________ minutes

## IV. COGNITIVE LOAD ASSESSMENT

Based on your experience, please rate the following statements:

0 = Very Low     ...      10 = Very High

1. **Mental Demand:** How much mental and cognitive effort was required to complete the task?
   0 [   ] 1 [   ] 2 [   ] 3 [   ] 4 [   ] 5 [   ] 6 [   ] 7 [   ] 8 [   ] 9 [   ] 10 [   ]
2. **Task Difficulty:** How difficult did you find the task?
   0 [   ] 1 [   ] 2 [   ] 3 [   ] 4 [   ] 5 [   ] 6 [   ] 7 [   ] 8 [   ] 9 [   ] 10 [   ]
3. **Stress Level:** How much stress or frustration did you feel while completing the task?
   0 [   ] 1 [   ] 2 [   ] 3 [   ] 4 [   ] 5 [   ] 6 [   ] 7 [   ] 8 [   ] 9 [   ] 10 [   ]

## V. TRUST AND WILLINGNESS

Based on your experience, please rate the following statements:

1 = Strongly Disagree     2 = Disagree     3 = Neutral     4 = Agree     5 = Strongly Agree

1. I believe the information I read/received is accurate.      1 [   ] 2 [   ] 3 [   ] 4 [   ] 5 [   ]
2. I am confident in my ability to understand and manage personal privacy.  1 [   ] 2 [   ] 3 [   ] 4 [   ] 5 [   ]

**(Group B only)**

3.c. I trust the answers provided by the AI agent.                    1 [   ] 2 [   ] 3 [   ] 4 [   ] 5 [   ]

4.d. I am willing to use AI agents in the future to help me understand other privacy policies.    1 [   ] 2 [   ] 3 [   ] 4 [   ] 5 [   ]

## VI. OPEN-ENDED FEEDBACK

1. What difficulties did you encounter during the reading/usage process?
2. How do you think the experience of reading privacy policies could be improved?

**(Group B only)**

3. Do you have any suggestions or comments regarding the AI agent?

