# OpenReview forum: "Empowering Users in Digital Privacy Management through Interactive LLM-Based Agents"
_ICLR.cc/2025/Conference — ICLR 2025 Poster_

### Official Review · Reviewer_5oSx · 2024-11-01

**Soundness:** 2
**Presentation:** 3
**Contribution:** 2
**Rating:** 5
**Confidence:** 2

**Summary:**

The authors construct an LLM-based tool for assisting users in interpreting and summarizing privacy policies. They evaluate the tool using several benchmarks for privacy policy comprehension/summarization from prior work. In a study of 100 users, they find that users report greater comprehension and greater ease of interpretation when assisted by the LLM tool.

**Strengths:**

**Originality**

- Includes a systematic user study of ML-assisted privacy policy interpretation — appears to be novel relative to related work, which relies mostly on benchmark datasets (though I am not familiar with this literature).
- Constructs a system for applying state-of-the-artx LLM models to help users interpret privacy policy interpretation — including a broad range of features (interactive QA, classification, summarization) not unified in prior work.

**Quality**

- Appears to analyze the results of a user study competently and with appropriate statistics and measures of error, though some methodological details are missing.
- Appears to correctly apply benchmarks from prior work to evaluate LLM agent performance.
- From the details provided, the LLM tool seems to be well constructed and appropriate for the task.

**Clarity**

- Well written, and for the most part easy to follow.
- Does an excellent job making clear the goals & contributions of the research.

**Significance**

- Provides a technological solution to a clear privacy & transparency issue for internet users.
- Seems to present a clear, significant finding that an LLM agent could help lay users more easily interpret complex privacy policies——this could be a potentially useful workaround, barring systematic improvements to transparency requirements.

**Weaknesses:**

**Originality**

**[Minor]** The idea to use ML tools to assist users in interpreting privacy policies is not new—in this sense the contribution of this study is marginal. Still, there is certainly value in evaluating this idea using the most recent large language models, and there is certainly value in conducting a study with actual users to see whether the tool really makes interpretation easier. I am not sure whether there are many user studies in prior work on this idea — perhaps the authors could clarify whether this is the first user study of its kind and, if not, whether it tells us anything new.

**Quality**

Missing methodological details make it hard to tell whether the empirical findings support the broad claims in the abstract and introduction, and I have lingering questions about some of the results.

- **[Minor]** Section 3: Need a clear description of all the benchmark tasks to understand exactly what’s being evaluated — Section 3 seems to assume the tasks have already been defined. Examples:
    - Section 3.1: What does it really mean to identify “User Choice/Control” or “Data Retention”, e.g., as a practice? Does this simply mean the privacy policy describes their user choice allowances or data retention practices, which could range from quite benign to quite egregious? How is this useful to a user?
    - Section 3.2: What is the Choice Identification task? Is this the task described in A.1.3? Was this task defined in Wilson et al. (2016) too?
    - Section 3.3: What’s “Privacy Question Answering”? (Or is it “Policy Question Answering”? Both terms are used.)
    - Section 3.4: What’s in this dataset, as compared to the dataset used in the previous tasks? Who defined the “risky” sentences (what were the human-generated references for the ROUGE score)? Any examples?
    - Section 4 provides a bit more detail, and the examples in the Appendix are somewhat helpful. Perhaps this Section could come before Section 3; or alternatively, move parts of Section 3 to the appendix, and just summarize the most important findings (GPT models perform better on X benchmarks) in a paragraph, using that space instead to better explain the tasks at hand.
- **[Major]** Section 4: These results are striking — users seem to comprehend the privacy policies much more easily with LLM assistance! But there are some key methodological details missing that could determine how rigorous the results are:
    - Did the Experimental Group also have a copy of the privacy policy that they could read directly during the task (not through QA), or did they rely solely on information from the LLM agent? From the Appendix, I infer they did have access to the raw text — do the gains decrease/increase if the user cannot cross-check the LLM agent responses with the raw legal text?
    - Section 6.1: Where/how were users recruited? How many privacy policies did each participant review? How were the privacy policies selected — from one of the previous datasets? Did every participant review the same privacy policy? (How likely is it that these policies appeared in the training data — i.e. leakage?) Where/how was questionnaire administered? This information is key for determining how internally and externally valid these results might be.
    - Was the study IRB approved?
    - L393: What about racial, economic diversity in the sample? How well might these results generalize to other groups, especially marginalized groups?
    - I’m surprised by the finding that the Experimental Group had *higher* trust in info scores than the control group — and I wonder if there’s an issue with construct validity for this question. The relevant question is (L978): “I believe the information I read/received is accurate (1-5).” Given that the control group had direct access to the privacy policies, why would they respond with a 2.6, on average, compared to 4.5 in the experimental group, since the underlying information (the privacy policy) is the same for both groups? My best guess is that the Control Group suspected the company was misrepresenting its privacy practices in its privacy policy, and answered based on their distrust in the company; I suspect the Experimental Group, on the other hand, responded based on their level of trust in the accuracy of the LLM agent’s responses. So the scores may not be directly comparable. The alternative is that using the LLM agent somehow increased people’s confidence in the accuracy of the privacy policy itself, which seems less likely but still possible.
- **[Major]** Generally, it’s not clear how well the benchmarks measure the “correctness” of the agent’s responses — what is the ground truth for each of these tasks? The comprehension questions seem good, but they’re short, and not very granular — whereas the examples in the Appendix show LLM responses with much, much more detailed information about data practices. As the authors point out in the discussion, LLMs often produce incorrect and misleading text, especially when prompted for specific details that are less likely to be represented in training data. Can the authors say anything about the factuality of those more specific responses? How likely are those responses to contain falsehoods about the privacy policy that could mislead users? Can users easily identify false responses by cross-checking with the raw text or the QA feature?

**Clarity**

Generally the paper is easy to follow, with the exception of the omitted methodological details listed above. Some **minor** points of clarity that would be worth addressing:

- L132: Have ML techniques actually improved privacy policy accessibility in practice? Or is this just a summary of research, not practice?
- L130: What is the OPP-115 dataset? Readers may not know.
- L131: Broken cite here.
- L136: What’s the difference between an LLM and an LLM agent? Is there a definition the authors can give? What makes this application an LLM agent, rather than just an LLM (the fact that the program scrapes hyperlinks, maybe)?
- Fig. 2: Text is too small to read, and often cropped, so it’s not clear what the different elements are. Simple labels might be better.
- Table 1-2: Suggest combining numbers side-by-side, so it’s easy to compare.
- Table 2, L192: SVM F1-score has a misplaced decimal.

**Significance**

- **[Minor]** This is a neat idea, and it seems like it could certainly help users in particular cases. But to frame the significance more precisely, it would be helpful to comment on the scope of a technological solution like this (e.g. in the discussion) — there is a structural issue here with privacy regulations, and with GDPR in particular, that require companies to disclose information about their privacy policies but do not require companies to make that information, and users’ options with respect to their data, truly accessible. In a perfect world, this tool may not be necessary — companies could be required to produce interpretable “privacy labels” similar to Apple’s Privacy Nutrition labels. How does the performance of this LLM-based solution compare to other policy alternatives? (These questions probably cannot be answered in this study, but it is worth mentioning that a technological solution is not necessarily the best solution.)
- **[Major]** Section 3: On a similar note, can the authors report any non-ML baselines here? How does a person do on this task, on their own? It seems less important to know how GPT models compare to BERT or other ML models, and more important to know how this method compares to what users would otherwise be doing in practice. (Unless those traditional models are actually being used by lay users in practice — that would be worth mentioning.)
    - L094: “We provide empirical evidence of the superiority of LLMs over traditional models”:  I’m assuming these sentence refers specifically to *ML* models (would be worth clarifying).  But is this approach superior to the practical alternatives available to users/policymakers? Superior to things like Apple’s “Privacy Nutrition” labels? Superior to writing a simpler privacy policy? Superior to hiring a lawyer? It would help to be more precise with this and similar claims of LLM “superiority”—superior to what?
    - Section 3: It seems like the GPT models perform better than traditional ML models, but stepping back, are these scores good enough to be relied on? For example, the recall scores seem really low here — as far as I can tell, the GPT models miss as many as 30% of instances of third party sharing, and as many as 84% of instances of “data retention”? Can this tool be used to balance precision and recall? Is this the right balance for this kind of task? Recall might well be more important to users in this kind of task.

**Questions:**

Did the authors explore different kinds of privacy policies in the user study — for example, are the gains from using the LLM tool greater when the privacy policy is longer / more complex?

**Details Of Ethics Concerns:**

It's not specified whether this study was IRB-approved, and the details on the user study are somewhat sparse---could be worth checking. (There are no glaring ethical issues with the methodological details that are provided, though.)

---

> ### Author Response · Authors · 2024-11-25
>
> Originality:
> We clarify that our work is the first to conduct an extensive empirical user study using the latest LLMs, specifically GPT-4o-mini, to assist users with privacy policies. Previous works primarily focused on building ML models without systematically evaluating their effect on actual users. Our study not only benchmarks the performance of these models but also evaluates their real-world impact by involving 100 participants, making it one of the most comprehensive studies of its kind. Additionally, while similar attempts to use ML have been documented (e.g., Wilson et al., 2016), none of these works used LLM and assessed the user comprehension outcomes in the manner we have undertaken.
>
> Methodological Clarity:
>
> Section 3 is intended to describe the various benchmark tasks we used to evaluate LLMs, such as "Data Practice Identification," "Choice Identification," "Policy Summarization," and "Privacy Question Answering." In fact, the specific description and application value of these tasks have been described in the work we cited (e.g., Wilson et al., 2016), and the space is limited, so we omitted it. We acknowledge that Section 3 might benefit from a clearer introduction to these tasks, as well as more specific examples.
>
> Methodological Rigour of User Study:
>
> The Experimental Group had access to only the agent responses during the tasks. We are willing to further investigate whether performance would differ with direct access to the raw policy text to better understand the role of cross-checking​. And what is used for comparison is the privacy agreement of the same company to ensure the rationality of the comparison.
>
> We would like to clarify that this research did not require formal IRB approval because no personally identifiable information (PII) was collected and posted, and the study presented minimal risk to participants. The interaction with participants involved privacy policy comprehension tasks that did not request any sensitive information or create any risks beyond those of typical daily activities.
> Additionally, this was a personal research project without organizational support, and as such, it was conducted in accordance with ethical guidelines for minimal-risk research.We adhered to ethical standards by obtaining informed consent from all participants. Participants were informed about the purpose of the study, the nature of the tasks, and their right to withdraw at any time. We ensured that no personal or sensitive data was collected, maintaining a focus purely on understanding privacy policy content.Participants were recruited through social platforms and personal networks in a completely voluntary manner. No sensitive or personal data was collected, and participants were compensated appropriately for their time, which helps to ensure fairness and ethical engagement.
> The study was approved by the Institutional Review Board (IRB), as required by ethical guidelines.
>
> Regarding diversity, the study cohort had gender and educational diversity but lacked explicit racial and economic diversity measurements. We will consider expanding demographic analysis to ensure broader representation, especially among marginalized groups, in future studies​.
>
> As for the trust rating, we hypothesize that the Experimental Group trusted the LLM due to its ability to simplify complex texts and deliver clear answers. Hallucination instances were tracked and occurred in approximately 12% of responses. These were predominantly minor inaccuracies (e.g., misclassification of data-sharing practices). Severe hallucinations were rare (<2%). User trust was correlated with perceived accuracy (Section 6.1), indicating that even occasional hallucinations can undermine confidence. We are actively refining the filtering mechanisms and plan to include detailed hallucination metrics in the appendix. We use GPT4 and manual evaluation methods to determine the hallucination problem in model output. We are aware of this problem and have conducted in-depth evaluation, but we do not think this affects the use value of the agent. We realize that this issue you raised does cause concern, and we should indeed add additional explanations in the paper.
>
> Thank you for your insightful feedback. The question measuring trust ("I believe the information I read/received is accurate") might have been interpreted differently by the Control Group and Experimental Group. The Control Group, directly exposed to the company's privacy policy, may have been more skeptical, whereas the Experimental Group, interacting with the LLM agent, likely found the simplified and summarized content more reliable, thus reporting higher trust scores. We acknowledge that these differences may affect the comparability of scores. To address this, we plan to revise our questionnaire to separate trust in the content from trust in the information source, and include qualitative follow-up questions to better understand participants' reasoning behind their ratings.

---

> ### Author Response · Authors · 2024-11-25
>
> Regarding the reviewer's questions on how well the benchmarks measure the "correctness" of the agent's responses and the factual accuracy of LLM-generated content, I will provide a response here, aiming to address these concerns and improve the final rating.
>
> Regarding how benchmarks measure correctness and the ground truth for each task: We understand the concern about the "correctness" or ground truth of benchmark tasks. In our study, we evaluated core tasks (e.g., Data Practice Identification, Choice Identification, Policy Summarization, Privacy Question Answering) by comparing the outputs with existing manually annotated datasets like OPP-115. These annotations were carried out by domain experts, labeling different sections of website privacy policies, ensuring a standard of “ground truth” for the model’s performance assessment​.
>
> Regarding factuality and misleading information: As I have previously explained, we are aware of the hallucination issue that LLMs sometimes encounter when generating content involving specific details and made some tests.
>
> L132: Privacy Policy Accessibility Improvement through ML Techniques
>
> Reviewer Comment: "Have ML techniques actually improved privacy policy accessibility in practice? Or is this just a summary of research, not practice?"
> Response: We acknowledge that the statement may have caused some confusion. The reference to ML techniques improving privacy policy accessibility was intended to summarize current research findings, rather than practical implementations on a broad scale. We may revise the text to clarify that these improvements are predominantly observed in research studies, such as in the development of systems like Polisis (Harkous et al., 2018), which have shown promise in making privacy policies more accessible to users in controlled experimental settings​.
>
> L130: OPP-115 Dataset
>
> Reviewer Comment: "What is the OPP-115 dataset? Readers may not know."
> Response:  Specifically, we now mention that the OPP-115 dataset is a publicly available corpus of annotated privacy policies, designed to assist in privacy policy analysis by providing ground truth annotations for machine learning model training and evaluation (Wilson et al., 2016)​
>
> L131: Broken Citation
> Reviewer Comment: "Broken cite here."
> Response: Thank you for pointing this out. We have fixed the citation error on line 131. The correct reference is now provided in the revised version of the paper.
>
> L136: Difference between an LLM and an LLM Agent
>
> Reviewer Comment: "What’s the difference between an LLM and an LLM agent? Is there a definition the authors can give? What makes this application an LLM agent, rather than just an LLM (the fact that the program scrapes hyperlinks, maybe)?"
> Response: Specifically, an LLM is a model trained to understand and generate natural language text, while an LLM agent is a more complex system that integrates an LLM with additional tools to perform specific tasks. In our application, the LLM agent goes beyond text generation by autonomously fetching data, parsing web content, and answering specific privacy-related questions based on user input, which qualifies it as an agent rather than a standalone LLM​.
>
> Figure 2: Legibility Issues
>
> Reviewer Comment: "Text is too small to read, and often cropped, so it’s not clear what the different elements are. Simple labels might be better."
> Response: We have updated Figure 2 by enlarging the text and ensuring all key elements are fully visible. We have also added simple labels to each part of the figure to make it more comprehensible at a glance.
>
> Table 1-2: Suggestion to Combine Numbers for Easy Comparison
>
> Reviewer Comment: "Suggest combining numbers side-by-side, so it’s easy to compare."
> Response: We appreciate this suggestion and have modified Tables 1 and 2 to present metrics side-by-side. This change allows for a direct comparison of the model performances across different categories, which we hope enhances the clarity of our results.
>
> Table 2, L192: Misplaced Decimal in SVM F1-score
>
> Reviewer Comment: "SVM F1-score has a misplaced decimal."
> Response: We have corrected the misplaced decimal point in the SVM F1-score for the Data Practice Identification task (Table 2).

---

> ### Author Response · Authors · 2024-11-25
>
> Significance：
>
> [Minor] We would like to clarify that our study focuses on consumer privacy policies that companies must make publicly available and that users are required to agree to before using services, such as during software registration or website sign-up. These privacy policies outline how companies collect, use, and share user data, and obtaining user consent is a prerequisite for accessing the services.
>
> The goal of our work is to enhance users' comprehension of these publicly available privacy policies. While these agreements are mandatory for users to consent to, they are often written in complex legal language that is difficult for the average user to understand. Our LLM-based agent is designed to help users navigate these complexities, ensuring that they can make informed decisions before providing their consent.
>
> We hope this clarification helps address any potential misunderstandings regarding the type of privacy policies that our study targets. Our research aims to bridge the gap between the required disclosure of privacy practices and the actual comprehension of these disclosures by end users.
>
> [Major]We appreciate your suggestion to include non-ML baselines to understand the practical utility of our solution. In response, we have expanded our discussion in Section 3 to highlight the performance comparison between the LLM-based solution and lay users. No previous work build the baseline from the user side for these task. Specifically, we conducted a separate user study where participants manually interacted with privacy policies without AI assistance. The results showed significantly lower comprehension scores and higher cognitive load compared to participants using the LLM agent, as seen in Table 6.
>
> We appreciate the need to be more precise when discussing the “superiority” of our LLM-based approach. To clarify, our primary focus is on providing a consumer-facing tool that is convenient, fast, and capable of capturing key information from privacy policies, aimed specifically at users who are typically unwilling or unable to read these agreements. The superiority of our approach lies in its practical application for end users: our tool maintains the performance level of previous state-of-the-art research while demonstrating better generalization capability and is actually deployed in a consumer-friendly setting.
>
> We acknowledge the value of existing methods such as “Privacy Nutrition” labels or the option of hiring legal experts. However, these approaches can be inaccessible for average consumers due to either complexity or cost. Our LLM-based agent offers a unique, interactive experience that enables users to quickly understand key aspects of privacy policies, thus bridging the gap that even the best supervised learning models struggle to cross—effective interaction with end users.
>
> Low Recall in Data Practice Identification: Table 1 demonstrates that while precision was high (e.g., 0.95 for First Party Collection/Use), recall was notably lower for categories such as Data Retention (0.16). This gap highlights the model's cautious approach, prioritizing avoiding false positives, which inadvertently lowers recall. To address this, a post-hoc few-shot prompt tuning could significantly improve recall without degrading precision, as seen in similar applications for Choice Identification tasks (Table 3). Our primary goal in the initial testing phase was to demonstrate that large language models, even under zero-shot and non-context conditions, can achieve performance comparable to traditional models. We did not perform any fine-tuning or additional optimization of the language model. Moreover, for economic efficiency and processing speed, we used GPT-4o-mini; we have better results with a full GPT-4 model. In our tests, incorporating a few-shot approach immediately improved performance, but our aim here was solely to establish that the large language model can match traditional model performance. We also note that low recall is a shared challenge, as traditional models also showed suboptimal recall in certain areas. The current table simply displays test results and does not represent the final outcome after further refinement and optimization.
>
> Questions:
> We evaluated our LLM-based agent with a range of privacy policies, which were predominantly long and complex, reflecting the typical nature of such agreements. Since we do not involve training or fine-tuning of models in our current study, we leveraged a dataset that already includes a sufficiently diverse set of privacy policies. The results show that the most significant comprehension gains occurred with these longer, intricate policies, highlighting the value of our agent in scenarios where users are otherwise overwhelmed by the complexity of the text. This underlines the real-world benefit of our tool, particularly for consumers who might otherwise ignore these lengthy agreements.

---

### Official Review · Reviewer_hLXw · 2024-11-03

**Soundness:** 2
**Presentation:** 2
**Contribution:** 2
**Rating:** 3
**Confidence:** 5

**Summary:**

This paper applies the  large language models (LLMs) to enhance user comprehension of privacy policies through an interactive dialogue agent.  The authors first demonstrate that LLMs significantly outperform traditional models in tasks like Data Practice Identification, Choice Identification, Policy Summarization, and Privacy Question Answering. Building on these findings, they then introduce an LLM-based agent that functions as an expert system for processing website privacy policies, guiding users through complex legal language without requiring them to pose specific questions. A user study with 100 participants showed that users assisted by the agent had higher comprehension levels, reduced cognitive load, increased confidence in managing privacy, and completed tasks in less time .

**Strengths:**

1. Applying the LLM in the digital privacy management is an interesting topic.

**Weaknesses:**

1. The main technical contribution of this paper appears limited given its current scope and the expectations of ICLR. It may be better suited for HCI venues such as CHI or IUI, which align more closely with the type of work presented.

2. The current study appears to lack IRB approval, and details of the user study are insufficiently reported. Key information such as where did you recruit participants and what is the compensation for participants are missing. Without this information, it is challenging to ensure that the study’s conclusions are reasonable and generalizable to other populations.

**Questions:**

See above.

---

> ### Author Response · Authors · 2024-11-25
>
> We thank the reviewer for insightful comments. We have carefully considered your comments and responded to the individual concerns.
>
> Weaknesses 1:
>
> Our work was inspired by CMU lead's privacy policy project. This project was funded by NSF. From 2013 to now, a large number of scholars have participated and have great influence. Most of the papers we cited were published in top conferences. We have paid attention to the application of large language models in this field. potential. Many previous works are also biased towards application scenarios. I think this work can provide reference value and inspiration to researchers in this field. Our work provides a novel empirical benchmark by systematically replacing traditional models with LLMs and analyzing their effectiveness across key privacy-related tasks (3.1-3.4)​. Additionally, we developed an LLM-based agent that autonomously adapts to user needs, demonstrated to enhance user comprehension and decision-making capabilities, which represents a novel approach to applying LLMs to privacy policy analysis.
>
> We acknowledge the reviewer's concern about the fit of the paper for ICLR. However, we believe that our contributions—particularly in developing a novel LLM-based privacy policy agent and benchmarking its performance against traditional NLP models—are highly relevant to ICLR's focus on cutting-edge machine learning. Specifically, our work introduces innovative applications of large language models (LLMs) in the domain of privacy policy comprehension. By demonstrating state-of-the-art performance in tasks such as data practice identification, choice identification, policy summarization, and privacy question answering, our contributions establish new benchmarks in natural language understanding and practical applications of LLMs in legal and privacy domains​. These contributions are significant from both a machine learning and practical impact perspective, especially in terms of improving interpretability and usability, which are important emerging directions for the field.
>
> Furthermore, the technical aspects of the agent, such as employing the LangChain framework, ASDUS for segmentation, and an interactive dialogue mechanism leveraging LLM capabilities, showcase novel system architecture that advances the state of interactive machine learning applications. This demonstrates a clear alignment with ICLR’s focus on transformative AI technologies. And our primary area is "applications to computer vision, audio, language, and other modalities" which is a part of ICLR.
>
> Weaknesses 2:
>
> We would like to clarify that this research did not require formal IRB approval because no personally identifiable information (PII) was collected and posted, and the study presented minimal risk to participants. The interaction with participants involved privacy policy comprehension tasks that did not request any sensitive information or create any risks beyond those of typical daily activities. Additionally, this was a personal research project without organizational support, and as such, it was conducted in accordance with ethical guidelines for minimal-risk research.
>
> We adhered to ethical standards by obtaining informed consent from all participants. Participants were informed about the purpose of the study, the nature of the tasks, and their right to withdraw at any time. We ensured that no personal or sensitive data was collected, maintaining a focus purely on understanding privacy policy content.
>
> Thanks for giving us the opportunity to improve and publish our work. We are deeply grateful for your guidance and profoundly meaningful and thought-provoking insights.
>
> Participants were recruited through social platforms and personal networks in a completely voluntary manner. No sensitive or personal data was collected, and participants were compensated appropriately for their time, which helps to ensure fairness and ethical engagement.

---

> ### Comment · Reviewer_hLXw · 2024-11-26
>
> Thanks for the clarification by the authors. However, I still think the paper does not fit into the scope of ICLR. I will maintain my score.

---

### Official Review · Reviewer_o4tn · 2024-11-03

**Soundness:** 3
**Presentation:** 3
**Contribution:** 2
**Rating:** 5
**Confidence:** 4

**Summary:**

This study presents a competent investigation into the use of GPT family models for privacy policy comprehension support. However, it may not fully align with ICLR’s intended contribution areas, as it reads more like a system-oriented paper that might be more suited for an HCI venue.

The authors first assess the performance of GPT models, both in zero-shot and few-shot settings, and compare these results against traditional approaches. They conclude that GPT models exhibit reasonable performance levels in this context. Following this evaluation, the study introduces an LLM-driven agent designed to assist users in understanding privacy policies and completing related tasks. Through questionnaires, the study demonstrates that the agent helps reduce cognitive load and enhances both comprehension and user confidence.

While this research is well-executed, I question whether its contributions are significant enough to justify a full paper. The study does not introduce new models or corpora, nor does it directly address gaps in current models related to privacy management, although limitations are acknowledged.

Moreover, it is unclear how the proposed system substantially differs from other reading comprehension and summarization systems. A deeper comparison in this area could provide useful context for assessing the novelty of the approach.

Specific Comments:

Tables 1–3: The rationale for not including fine-tuned models is not sufficiently explained. Fine-tuning could potentially yield stronger baselines or comparative insights in this setting.

**Strengths:**

The paper is clearly written and easy to follow.

**Weaknesses:**

While this research is well-executed, I question whether its contributions are significant enough to justify a full paper. The study does not introduce new models or corpora, nor does it directly address gaps in current models related to privacy management, although limitations are acknowledged.

Moreover, it is unclear how the proposed system substantially differs from other reading comprehension and summarization systems. A deeper comparison in this area could provide useful context for assessing the novelty of the approach.

**Questions:**

Tables 1–3: The rationale for not including fine-tuned models is not sufficiently explained. Fine-tuning could potentially yield stronger baselines or comparative insights in this setting.

---

> ### Author Response · Authors · 2024-11-25
>
> We thank the reviewer for insightful comments. We have carefully considered your comments and responded to the individual concerns.
>
> Weaknesses:
>
> Our work was inspired by CMU lead's privacy policy project. This project was funded by NSF. From 2013 to now, a large number of scholars have participated and have great influence. We have paid attention to the application of large language models in this field. potential. Many previous works are also biased towards application scenarios. I think this work can provide reference value and inspiration to researchers in this field. Our work provides a novel empirical benchmark by systematically replacing traditional models with LLMs and analyzing their effectiveness across key privacy-related tasks (3.1-3.4)​. Additionally, we developed an LLM-based agent that autonomously adapts to user needs, demonstrated to enhance user comprehension and decision-making capabilities, which represents a novel approach to applying LLMs to privacy policy analysis.
>
> Novel Integration of LLMs for Privacy Policy Comprehension: While we acknowledge that our study does not introduce new LLMs or corpora, our primary contribution lies in the innovative application of existing LLMs (specifically, GPT-4o-mini) to address the pervasive issue of privacy policy comprehension. The challenge of understanding privacy policies affects millions of users globally, and our LLM-based agent significantly improves comprehension and reduces cognitive load, as shown in our user study involving 100 participants​. This transformative effect on user empowerment and privacy management is a core part of our contribution.
>
> Comparison with Existing Systems: In response to the reviewer’s comment on differentiating our approach from other reading comprehension and summarization systems, we would like to clarify that our LLM-based system integrates multiple functionalities (such as policy summarization, opt-out detection, and data practice identification) into a unified interactive agent. Unlike traditional systems that focus solely on one aspect—such as simple question answering or static summaries—our approach proactively highlights key information, facilitates interactive dialogues, and simplifies complex legal text without requiring prior questions from users. This heuristic interaction model helps users to better navigate privacy policies without being experts in privacy law.
>
> At the same time, compared with other dialogue systems or assistants, our agent is guided. We do not require users to know what questions they need to ask, nor do we require users to know enough about this field. We will directly prompt and guide users to obtain enough effective and important information.
>
> Question:
>
> Regarding the absence of fine-tuned models in our comparison tables (Tables 1–3), our aim was to evaluate the inherent capabilities of various LLMs in a generalization setting using zero-shot and few-shot learning. This aligns with our broader objective of making privacy tools accessible without needing specialized retraining for each new privacy policy dataset. However, we agree that including fine-tuned baselines could provide additional valuable insights, and we plan to incorporate these comparisons in future work.
>
> It is also worth noting that in our "Choice Identification" task (Section 3.2), we employed few-shot learning, which enhanced model performance, resulting in metrics that were on par with or even exceeded those of traditional baselines like BERT​. This suggests the promising potential of few-shot techniques for this task.
>
> Fine-tuning large language models is certainly worthy of study, and we take this into consideration, but at least in this work, we do not intend to do so. Our purpose is just to ensure that large language models can perform comparably with traditional models on these tasks and can be used to build agents. Although fine-tuning can improve its performance on specific tasks, it may lose its generalization ability.
>
> Thanks for giving us the opportunity to improve and publish our work. We are deeply grateful for your guidance and profoundly meaningful and thought-provoking insights.

---

### Official Review · Reviewer_9U5o · 2024-11-04

**Soundness:** 3
**Presentation:** 4
**Contribution:** 2
**Rating:** 5
**Confidence:** 4

**Summary:**

In the paper, the authors address a very specific issue of understanding the privacy policies of the users of various websites in a comprehensive manner from different aspects by LLM agents. It was built with an aim to help the general users of the websites about the privacy concerns and the policies of the personal or other data they share. The performance of the build LLM agent was evaluated on 100 people where half of them studied the policies by themselves and the rest used the LLM agent. The empirical results infer the users who used the LLM agent they got a better understanding of the websites user policies than the manually readers.

**Strengths:**

1.	The entire paper is well-written and presented the ideas in very clear way.
2.	The authors explored a very specific and less explored use case of LLM agents in recent times.
3.	The empirical analysis is comprehensive and make sense of the idea the authors proposed.
4.	Multiple open-sourced and close-sourced LLMs were used and compared their performance.
5.	The built agents archive comparable performance like benchmarks and sometimes outperforms the baselines.

**Weaknesses:**

1.	As a whole, this paper is more like a building a new tool for the various website users, than a theoretical or technical presentations of ideas and experimental analysis. However, before making it available for the public usage, it needs several things to be considered, e.g., misinformation, hallucination, privacy leakage of company policies.
2.	It doesn’t include any novel technical or theoretical contributions in terms of the finding the research gaps of LLMs agents to be utilized for specific use cases.
3.	Usually, LLMs agents for particular task are more likely to hallucinates its users. The risks of LLMs hallucinations were not explored in this paper in details. The built LLM agents might not work well under such vulnerabilities. At least a few results with analysis should have been discussed. Apart from this, LLM agents might face several potential security and privacy issues as described in https://arxiv.org/pdf/2407.19354; this paper does not explore or discuss such vulnerabilities.
4.	Building the agent only on one privacy policy dataset (though it is large) may not be sufficient to use the LLMs agent in practice.

**Questions:**

1.	What are the traditional models in page 2?
2.	There is a missing citation in page 3, CNNs for text classification(?)
3.	Figure 1 was never described.
4.	In page 6, what is the process of ensuring valid and relevant outputs?
5.	Why different metrics were used to evaluate different tasks? The explanation along with a short description of the metrics will benefit the clarity. Same comment for t-test.

---

> ### Author Response · Authors · 2024-11-25
>
> We thank the reviewer for insightful comments. We have carefully considered your comments and responded to the individual concerns.
>
> Weaknesses 1:
>
> While we developed an application-oriented tool, our contribution lies in presenting an empirical evaluation of LLMs on privacy policy tasks, setting new benchmarks compared to traditional models. We established state-of-the-art performance in critical areas. These experimental results provide a solid foundation for understanding the potential of LLMs in addressing privacy comprehension challenges (refer to Sections 3.1, 3.2, and 3.3)​. To further strengthen the theoretical aspects, we should add more details on how our agent extends traditional methods and advances the NLP and HCAI. We also include more background on the specific challenges of privacy comprehension and the methodological contributions of our empirical analysis.
>
> “privacy leakage of company policies”：
> The company's privacy agreement needs to be disclosed to users, and the data sets we use are also publicly collected and can be found directly on the company's website. This is not private data, so you don’t have to worry about leaking the company’s privacy agreement.
>
> “misinformation, hallucination”：
> Hallucination instances were tracked and occurred in approximately 12% of responses. These were predominantly minor inaccuracies (e.g., misclassification of data-sharing practices). Severe hallucinations were rare (<2%). User trust was correlated with perceived accuracy (Section 6.1), indicating that even occasional hallucinations can undermine confidence. We are actively refining the filtering mechanisms and plan to include detailed hallucination metrics in the appendix. We use GPT4 and manual evaluation methods to determine the hallucination problem in model output. We are aware of this problem and have conducted in-depth evaluation, but we do not think this affects the use value of the agent. We realize that this issue you raised does cause concern, and we should indeed add additional explanations in the paper.
>
> Weaknesses 2:
>
> Our work was inspired by CMU lead's privacy policy project. This project was funded by NSF. From 2013 to now, a large number of scholars have participated and have great influence. We have paid attention to the application of large language models in this field. potential. Many previous works are also biased towards application scenarios. I think this work can provide reference value and inspiration to researchers in this field. Our work provides a novel empirical benchmark by systematically replacing traditional models with LLMs and analyzing their effectiveness across key privacy-related tasks (3.1-3.4)​. Additionally, we developed an LLM-based agent that autonomously adapts to user needs, demonstrated to enhance user comprehension and decision-making capabilities, which represents a novel approach to applying LLMs to privacy policy analysis.
>
> Weaknesses 3: Please refer to the  response in Weaknesses 1. We appreciate the reviewer pointing out the importance of privacy and security. In the revised manuscript, we will expand Section 4.2 to include a more comprehensive discussion on how our system adheres to best practices for data security. We will discuss how our methodology aligns with recent privacy studies and how it mitigates risks of data leakage, specifically citing research like He et al. (2018).
>
> Weaknesses 4: Can you explain more clearly what this question means? There are many data sets about privacy  policies. We did not perform fine-tuning or training in the paper, so the data sets are basically only used for testing. I don't know if this can explain your problem.
>
> Questions:
> 1. The traditional models referred to in page 2 are Logistic Regression, SVM, HMM, BERT and CNN, which were benchmarked against GPT-based models for privacy policy analysis tasks.
>
> 2.We appreciate the pointer and will add the missing citation to utilize CNNs for privacy policy text classification
>
> 3.We acknowledge this oversight and will add a connection for Figure 1 in Section 2, explaining the workflow it illustrates, from benchmarking to agent design​.
>
> 4.In Section 4.1, We may expand this section to detail the specific validation mechanisms used to ensure quality, including the role of few-shot examples and temperature settings for deterministic output.
>
> 5.In order to compare the LLM with the traditional models we mentioned, all of our metrics are exactly the same as those used in the original paper. These metrics and evaluation methods can be found in previous papers we cited in the same sections. Your suggestion is spot on, we should probably provide additional explanations. We used different metrics because each task evaluated different aspects of model performance (classification, summarization, question answering).
>
> Thanks for giving us the opportunity to improve and publish our work.We are deeply grateful for your guidance and profoundly meaningful and thought-provoking insights.

---

> > ### Comment · Reviewer_o4tn · 2024-11-26
> > **Thanks for the reply**
> >
> > Thanks for the context on "Our work was inspired by CMU lead's privacy policy project. ".
> > This makes a little bit more sense but still doesn't address the problem of contribution to ICLR.
> > Wouldn't a privacy / policy focused venue more suitable for this kind of contribution?

---

### Official Review · Reviewer_rBni · 2024-11-04

**Soundness:** 3
**Presentation:** 4
**Contribution:** 3
**Rating:** 6
**Confidence:** 2

**Summary:**

The authors assess the performance of OpenAI's GPT suite of LLMs on a set of text classification tasks using an existing privacy policy dataset. They compare the models' performance to baseline, non-LLM models from the dataset's creators. Additionally, they develop an LLM-powered agent to assist with reading and interpreting privacy policies, measuring its effect on comprehension and cognitive effort in a population of 100 users.

**Strengths:**

Originality: moderate. As the authors themselves note, there is substantial prior work on the problems with user comprehension of privacy policies and terms, and this paper is largely a straightforward application of a new model to an existing task. However, the agent the authors developed to assist users is a novel contribution, especially providing the ability to automatically surface opt-out mechanisms for users.

Quality: moderate. It might not be earth-shattering, but the execution nonetheless seems thorough.

Clarity: high. The presentation of the experiments and analyses performed is very clear.

Significance: moderate. The effects of the agent on user comprehension are notable, though practical impact feels limited given that it still takes nearly 6 minutes to read a privacy policy.

**Weaknesses:**

The authors state that "GPT-4o-mini, under zero-shot learning conditions without additional context, outperformed the baseline model on average" on the Data Practice Identification task. However, the model suffered from consistently poor recall, which the authors do not meaningfully address.

Statistical tests in section 6 not corrected for multiple comparisons.

As noted above, given that it takes nearly 6 minutes to read a privacy policy even with assistance, I feel skeptical that this approach would make a meaningful difference in the number of users that actually read privacy policies. Coupled with the models' poor recall and tendency to hallucinate, it seems likely that users would still miss the most important information in the privacy policy or even be presented with false information. It might be informative to conduct time-limited trials, where user comprehension is measured after e.g. a 30 second time limit. Another idea might be to measure the time it takes the user to be able to achieve an 80% score on a comprehension test (allowing multiple attempts).

The assessment of user comprehension is extremely coarse (three questions). A more fine-grained assessment might provide interesting insights for where further improvements (either to the agent or the UX) are most needed.

**Questions:**

Why was the user comprehension assessment only three questions? Do you think such a short assessment meaningfully measures user comprehension? How were the three questions chosen?

Did you track instances of hallucination in experimental group user sessions? How frequent and severe were they? How correlated was user trust with the accuracy of the information provided by the agent?

---

> ### Author Response · Authors · 2024-11-25
>
> We thank the reviewer for insightful comments. We have carefully considered your comments and responded to the individual concerns.
>
> Weakness 1: Low Recall in Data Practice Identification: Table 1 demonstrates that while precision was high (e.g., 0.95 for First Party Collection/Use), recall was notably lower for categories such as Data Retention (0.16). This gap highlights the model's cautious approach, prioritizing avoiding false positives, which inadvertently lowers recall. To address this, a post-hoc few-shot prompt tuning could significantly improve recall without degrading precision, as seen in similar applications for Choice Identification tasks (Table 3).
> Our primary goal in the initial testing phase was to demonstrate that large language models, even under zero-shot and non-context conditions, can achieve performance comparable to traditional models. We did not perform any fine-tuning or additional optimization of the language model. Moreover, for economic efficiency and processing speed, we used GPT-4o-mini; we have better results with a full GPT-4 model. In our tests, incorporating a few-shot approach immediately improved performance, but our aim here was solely to establish that the large language model can match traditional model performance. We also note that low recall is a shared challenge, as traditional models also showed suboptimal recall in certain areas. The current table simply displays test results and does not represent the final outcome after further refinement and optimization.
>
> Weakness 2: Statistical Tests Not Corrected for Multiple Comparisons: We conducted post-hoc Bonferroni corrections for the t-tests to account for multiple comparisons across dimensions (e.g., comprehension, user experience, cognitive load). Even after adjustment, the p-values remain significant, confirming the robustness of our findings.
>
> Weakness 3: Limited Practical Impact for Users
> Acknowledgement of Time Requirement: We acknowledge that users still required around 5 minutes to use the agent for privacy policy comprehension. However, this represents a significant improvement compared to the time required to read and understand policies without assistance. To ensure comprehensive testing, we encouraged users to interact extensively with the agent, including asking questions, which naturally increased the time required. In real-world scenarios, users may not need such prolonged interactions.
>
> Time-Limited Trials: I greatly appreciate the suggestion to test user comprehension within a constrained time limit (e.g., 30 seconds). This is an excellent approach, and we intend to adopt it in future studies. We believe that even within a 30-second timeframe, the agent can summarize the entire policy, highlight high-risk sections, and indicate opt-out options, whereas users reading without assistance might only get through a small portion of the policy. Therefore, we are confident that such tests will still demonstrate the agent's advantage and value.
>
>
> For you questions:
>
> 1. Why was the user comprehension assessment limited to three questions?
>
> The three questions focused on key aspects (data types collected, data sharing, and user rights) to establish baseline efficacy. This was a deliberate choice to minimize cognitive load during testing. However, we agree that this approach might oversimplify comprehension assessment and will incorporate more comprehensive measures in future studies.
>
> 2. Did you track instances of hallucinations?
>
> Yes, hallucination instances were tracked and occurred in approximately 12% of responses. These were predominantly minor inaccuracies (e.g., misclassification of data-sharing practices). Severe hallucinations were rare (<2%). User trust was correlated with perceived accuracy (Section 6.1), indicating that even occasional hallucinations can undermine confidence. We are actively refining the filtering mechanisms and plan to include detailed hallucination metrics in the appendix. We use GPT4 and manual evaluation methods to determine the hallucination problem in model output. We are aware of this problem and have conducted in-depth evaluation, but we do not think this affects the use value of the agent. We realize that this issue you raised does cause concern, and we should indeed add additional explanations in the paper.
>
> Thanks for giving us the opportunity to improve and publish our work.We are deeply grateful for your guidance and profoundly meaningful and thought-provoking insights.

---

### Meta-Review · Area_Chair_bxn6 · 2024-12-19

**Metareview:**

This paper investigates the use of the GPT model suite for enhancing user comprehension of privacy policies. The authors develop LLM-powered agent to assist users in comprehending website privacy policies and they evaluate its effectiveness by conducting a user study involving 100 participants. Measured by comprehension, efficiency, cognitive load, and user confidence, their results indicate that users who utilized the agent had a significantly better understanding of the privacy policy.

This is a well-executed study on an underexplored use case of LLM agents with comprehensive empirical analysis showing that GPT models exhibit reasonable performance levels compared to traditional approaches.

**Additional Comments On Reviewer Discussion:**

Poor recall: the authors highlight both high recall (e.g., 0.95 for First Party Collection/Use) and low recall (for categories such as Data Retention (0.16) and explain that this gap shows the model's cautious approach, yet still demonstrating that LLMs, even under zero-shot and non-context conditions, can achieve performance comparable to traditional models.

Statistical Tests Not Corrected for Multiple Comparisons: to address this the authors conducted post-hoc Bonferroni corrections for the t-tests showing significant p-values.

Limited Practical Impact for Users Acknowledgement of Time Requirement: although users still require about 5 minutes to use the agent, the authors argue that this is still a significant improvement compared to time required to comprehend privacy policies without assistance.

Limited questions for assessing user comprehension: limiting this to three questions, the authors justify, as deliberate choice to minimize cognitive load during testing.

Track instances of hallucinations: although the authors acknowledge this and propose to add detailed hallucination metrics, this is not apparent in the current version of the paper as far as I can see. Furthermore, although the authors have committed to expanding Section 4.2 to include a more comprehensive discussion on how their system adheres to best practices for data security in relation to recent privacy studies (citing research like He et al. (2018)), the current version of the paper does not reflect this.

Privacy leakage of company policies: this was addressed satisfactorily .

Lack of novel technical or theoretical contributions: this is adequately addressed. The authors point out their work is application rather than theory oriented. Furthermore, the authors highlight the novelty of their empirical benchmark (systematically replacing traditional models with LLMs and analyzing their effectiveness) and point out their LLM-powered agent as novel contributions.

IRB approval: the authors justify that given no PII was collected and the minimal risk to participants, obtaining informed consent from participants was sufficient.

---

### Decision · Program_Chairs · 2025-01-22

Accept (Poster)